# PKA at a Cross-Road of Signaling Pathways Involved in the Regulation of Glioblastoma Migration and Invasion by the Neuropeptides VIP and PACAP

**DOI:** 10.3390/cancers11010123

**Published:** 2019-01-21

**Authors:** Souheyla Bensalma, Soumaya Turpault, Annie-Claire Balandre, Madryssa De Boisvilliers, Afsaneh Gaillard, Corinne Chadéneau, Jean-Marc Muller

**Affiliations:** 1Team Récepteurs, Régulations, Cellules Tumorales (2RCT), EA3842 CAPTuR, Pôle Biologie-Santé, Université de Poitiers, F-86022 Poitiers, France; souheyla.bensalma@gmail.com (S.B.); soumaya.turpault@gmail.com (S.T.); m.de.boisvilliers@outlook.fr (M.D.B.); corinne.chadeneau@univ-poitiers.fr (C.C.); 2STIM Laboratory, CNRS ERL 7003-EA7349, Pôle Biologie-Santé, Université de Poitiers, F-86022 Poitiers, France; annie-claire.balandre@univ-poitiers.fr; 3Laboratoire de Neurosciences Expérimentales et Cliniques (LNEC)–INSERM UMR-S1084, Pôle Biologie-Santé, Université de Poitiers, F-86022 Poitiers, France; afsaneh.gaillard@univ-poitiers.fr

**Keywords:** glioblastoma, protein kinase A, Sonic Hedgehog, Akt, PTEN, VIP-receptor system

## Abstract

Glioblastoma (GBM) remains an incurable disease, mainly due to the high migration and invasion potency of GBM cells inside the brain. PI3K/Akt, Sonic Hedgehog (SHH), and PKA pathways play major regulatory roles in the progression of GBM. The vasoactive intestinal peptide (VIP) family of neuropeptides and their receptors, referred in this article as the “VIP-receptor system”, has been reported to regulate proliferation, differentiation, and migration in a number of tumor cell types and more particularly in GBM cells. These neuropeptides are potent activators of the cAMP/PKA pathway. The present study aimed to investigate the cross-talks between the above cited signaling cascades. Regulation by VIP-related neuropeptides of GBM migration and invasion was evaluated ex vivo in rat brain slices explanted in culture. Effects of different combinations of VIP-related neuropeptides and of pharmacological and siRNA inhibitors of PKA, Akt, and of the SHH/GLI1 pathways were tested on GBM migration rat C6 and human U87 GBM cell lines using the wound-healing technique. Quantification of nuclear GLI1, phospho-Akt, and phospho-PTEN was assessed by western-immunoblotting. The VIP-receptor system agonists VIP and PACAP-38 significantly reduced C6 cells invasion in the rat brain parenchyma ex vivo, and C6 and U87 migration in vitro. A VIP-receptor system antagonist, VIP_10-28_ increased C6 cell invasion in the rat brain parenchyma ex vivo, and C6 and migration in vitro. These effects on cell migration were abolished by selective inhibitors of the PI3K/Akt and of the SHH pathways. Furthermore, VIP and PACAP-38 reduced the expression of nuclear GLI1 while VIP_10-28_ increased this expression. Selective inhibitors of Akt and PKA abolished VIP, PACAP-38, and VIP_10-28_ effects on nuclear GLI1 expression in C6 cells. PACAP-38 induced a time-dependent inhibition of phospho-Akt expression and an increased phosphorylation of PTEN in C6 cells. All together, these data indicate that triggering the VIP-receptor system reduces migration and invasion in GBM cells through a PKA-dependent blockade of the PI3K/Akt and of the SHH/GLI1 pathways. Therefore, the VIP-receptor system displays anti-oncogenic properties in GBM cells and PKA is a central core in this process.

## 1. Introduction

Glioblastoma multiforme (GBM) are the most frequent and malignant adult brain cancers. Their origin remains unclear, but it has been proposed that they could develop from the transformation of poorly differentiated glial progenitors or of neural stem cells [1,2,3,4]. Since 2005, the Stupp protocol has become the standard of care for the treatment of GBM. It consists of radiotherapy and concomitant chemotherapy with temozolomide, an alkylating agent [5]. Since then, in spite of constant progress of neurosurgical resection, radio- and chemotherapy, GBM remains an incurable disease with a median survival time less than two years after diagnosis. This is the consequence of the high migration and invasion potency of GBM cells, more particularly of the subset of so-called GBM stem cells or GSCs [6]. A strategy to cure GBM patients could thus be to inhibit tumor infiltration into the surrounding brain parenchyma. Clinical trials for GBM infiltration using the matrix metalloproteinase inhibitor *Marimastat* or the integrin antagonist *Cilengitide* have been attempted with no real success [7]. Numerous recent therapeutic trials targeting the pro-invasive role in GBM of Ephrin receptors, TGFβR1, Integrin β8 chain, Rho GTPases, and casein kinase 2 (CK2) are under development [8]. Recent immunotherapy early phase trials targeting the GBM stem cells led to a significant improvement of the median survival of patients [9].

The signaling pathways that play central roles in the invasive potential and in the radio- and chemo-resistance of GBM have been extensively studied. Among them are the PI3K/Akt/PTEN/mTOR and the SHH/GLI1 cascades [10]. In numerous GBM cases, PI3K/Akt is abnormally activated, due to amplification of EGFR, gene amplification, or activating mutations of the p110α catalytic or of the p85 regulatory subunits of PI3K. Almost half of GBM patients bear deletions, mutations, or epigenetic silencing of the PTEN gene leading to a loss of function of this anti-oncogenic factor associated with poor survival. Alterations of at least one of the EGFR, PTEN, or p110 PI3K genes is frequently detected in primary and or secondary GBM [11,12]. Effectors of this pathway have been targeted by a number of small molecules that demonstrated poor therapeutic benefit on GBM progression in clinical trials [13,14,15,16,17].

Another major cascade in GBM pathogenesis is triggered by the developmental protein Sonic Hedgehog (SHH) binding to the transmembrane glycoprotein Patched-1 (PTCH1), which releases its repressor activity on the smoothened (SMO) co-receptor, a member of the G-protein coupled receptors (GPCR) family. This causes the expression, activation, and nuclear import of glioma-associated oncogene homolog 1 (GLI1), a zinc finger transcription factor, regulating directly or indirectly the expression of numerous factors involved in GBM progression. Growth factors also activate GLI1 through the PI3K/Akt and Ras/MAP kinases cascades, while GPCR activation of PKA represses this process [18,19]. A number of small compounds that inhibit different effectors of this pathway have been developed. Despite their efficacy in vitro and in preclinical assays, SMO inhibitors like the plant alkaloid cyclopamine and its derivatives failed to improve the overall patient survival in clinical trials. This may be due to their limited bioavailability and to unintentional side effects, since the SHH pathway is involved in many physiological cell processes. Moreover, resistance to these inhibitors have been observed in animal models as a consequence of, for example, SMO activating or PTCH1 inactivating mutations, and PTCH1 suppression by the microRNA miR-9 [20,21,22,23,24,25].

The “VIP-receptor system” is composed of the 28-amino-acid neuropeptide VIP (vasoactive intestinal peptide) and VIP-related peptides, such as the 38-amino-acid PACAP-38 (pituitary adenylate-cyclase activating peptide) and their GPCR: VPAC1 and VPAC2, which display a high affinity for both VIP and PACAP-38, and PAC1 which is selective for PACAP-38. The pleiotropic functions of the VIP-receptor system in the body, particularly on glial and neuronal differentiation and on the progression of a number of cancer types, are at least partly mediated by a potent activation of the cAMP/PKA pathway [26,27,28,29,30]. We and others demonstrated that GBM generally express different combinations of components of the VIP-receptor system that are involved in the control of proliferation and migration of GBM cells [31,32,33,34,35,36,37,38]. VIP and PACAP act as anti-invasive factors in different GBM cell lines, a function mediated by VPAC1-dependent inhibition of AKT phosphorylation [36,38]. PACAP also acts as a strong tumor suppressor in medulloblastoma (MB), a highly aggressive tumor of the cerebellum, by repressing GLI1 expression and activity in a PKA-dependent manner [39,40]. It has been proposed that in MB, a compartmentalized pool of PKA in the vicinity of primary cilia could be involved in the regulation by PACAP of the SHH/GLI1 pathway [41]. PKA directly regulates GLI1 by phosphorylating a Threonine residue near the nuclear localization signal of this protein, resulting in its cytoplasmic retention [42]. Overexpression of Gαs in transgenic mice leads to increased cAMP-dependent signaling, inhibits ciliary trafficking of SHH components and SMO activation, leading to decreased MB cell proliferation [43]. These data indicate that, as an alternative to SMO inhibition, the SHH signaling could be inhibited by PACAP/PKA signaling in tumor cells.

Shortly after the discovery of the intracellular messenger cyclic adenosine monophosphate (cAMP) and of its synthetase adenylate cyclase (ADCY), a relationship between the levels of this nucleotide, cell differentiation and tumor biology has been evident. Decreased activation of the cAMP-dependent pathway and lower levels of cAMP are generally associated with increased tumor grade and decreased overall patient survival. An extensive and converging documentation supports the potent tumor suppressor function of the cAMP/PKA axis in GBM and MB cells [43,44,45,46,47,48].

All together, these observations encouraged us to investigate further the signaling pathways triggered by the VIP-receptor system involved in GBM migration and invasion, paying particular attention to the potential implication of PKA in the cross-talks between the PI3K/PTEN and the SHH/GLI1 cascades in these phenomena.

## 2. Results

### 2.1. Components of the VIP-Receptor System Are Expressed in C6 and U87 Cell Lines

Analysis by RT-qPCR of mRNA expression of the components of the VIP-receptor system reveals that C6 cells mainly express VIP and the VPAC2 receptor and very limited levels of PACAP, VPAC1 and PAC1 mRNAs. In U87 cells, the main expressed components are VIP and PAC1 with a very low level of VPAC1 and PACAP mRNAs, while VPAC2 mRNA was undetectable, as shown in Appendix A. Thus, both cell lines expressed mainly the VIP mRNA and at least one mRNA encoding a receptor of the VIP-receptor system.

### 2.2. VIP, PACAP-38, and VIP_10-28_ Effects on Invasion by C6-GFP Cells of the Rat Brain Parenchyma

Ex vivo invasion assays on rat brain slices were performed in order to assess the potential action of VIP and related peptides on this process in rat C6 GBM cells. After 48 h of treatment of the slices with 1 µM VIP, PACAP-38, or VIP_10-28_, the width of invasion of GFP-C6 cells was determined using a fluorescent macroscope, as shown in Figure 1A,B. The depth of invasion of GFP-C6 cells was assessed by confocal microscopy, as shown in Figure 1C. The width of the invasive area was significantly reduced by about 50% by the VIP or PACAP-38 receptors agonists and the depth of invasion was decreased by about 35%. On the contrary, the VIP receptor antagonist VIP_10-28_ increased by 2.3-fold the width of the invasive area and by about 1.4-fold the width of invasion by C6 cells. The VIP and PACAP-38 analog secretin, which does not interact with VIP/PACAP receptors, did not significantly modify invasion of brain slices by C6 cells, as shown in Figure 1B,C.

### 2.3. VIP, PACAP-38, and VIP_10-28_ Regulation of the Migration Process of C6 and U87 Cells is Affected by Different Signaling Pathways Inhibitors

Neuropeptide effects on GBM cell migration were evaluated by the wound-healing technique, as described in the methods section and as previously reported [36]. In both C6 and U87 cells, VIP or PACAP-38 but not secretin, dose-dependently and robustly reduced the wound-healing closure process, as shown in Figure 2A,B. The VIP receptor antagonist VIP_10-28_ significantly increased the migration of C6 cells but had no significant effect in the U87 cell line. The PKA inhibitor H89 at 1 µM abolished the inhibition of migration induced by 10^−7^ M VIP or PACAP-38 in both cell types, as shown in Figure 2C,D.

The increased migration of C6 cells by VIP_10-28_ was blocked by 1 µM of the AKT protein kinase inhibitor or by the SMO co-receptor antagonist cyclopamine. None of these inhibitors affected the cell migration on their own at the concentration used, as shown in Figure 3A. When C6 cells were transfected with a siRNA targeting GLI1 (Gli-1 siRNA), an important reduction of wound closure by about 3-fold was obtained when compared with cells transfected with a control siRNA. The VIP-receptors antagonist VIP_10-28_ significantly increased migration in the cells transfected with the control siRNA, but this effect could not be observed in the cells transfected with the GLI1-siRNA, as shown in Figure 3B. These results indicate that PKA is implicated in the regulation of C6 and U87 cell migration by the VIP-receptor system and that the component of the Hh pathway GLI1 participates also in this regulation in C6 cells.

### 2.4. Inhibitors of PKA (H89) or of Akt (IA) Abolish the Effects of VIP-Related Peptides on the GLI1 Protein Nuclear Expression

Western immunoblotting detection of GLI1 in C6 or U87 lineages revealed the major expression of this protein in nuclear extracts of these GBM cells, as shown in Figure 4A. Expression of GLI1 was analyzed in cells treated for 24 h with neuropeptides at 10^−7^ M, following or not a 1 µM pretreatment with H89 or IA, as shown in Figure 4B–D. In both cell types, VIP or PACAP-38 induced an important reduction of nuclear expression of Gli1 protein. The effects of these neuropeptides were totally blocked by H89. In C6 cells, the VIP_10-28_ antagonist triggered a robust rise of the expression of nuclear GLI1 protein, an effect which was completely abolished by IA.

### 2.5. PACAP-38 Triggers a Time-Dependent Decrease of Phospho-Akt and Elevation of Phospho-PTEN Protein Expression in C6 Cells

C6 cells were treated with 10^−7^ M PACAP-38 for increasing periods of time ranging from 30 min to 24 h. Western immunoblotting detection of total and phosphorylated forms of Akt and phosphorylated PTEN was then carried out in cell lysates from these experiments, as shown in Figure 5A–C. PACAP-38 time-dependently decreased the phospho-Akt/total Akt ratio with a very strong but transient maximal effect after 6 h of treatment. At 24 h, this ratio returned to its initial level at the onset of the experiment. On the opposite, the phospho-PTEN/GAPDH ratio started to significantly increase after 2 h of treatment with a maximal increase of about 10-fold after 3 h. Then this ratio dropped back to its initial value after 24 h. Increased phosphorylation of PTEN by PACAP-38 was abolished in the presence of the PKA inhibitor H89, as shown in Figure 5D,E.

## 3. Discussion

Previous work from our group demonstrated that the VIP-receptor system displayed anti-oncogenic properties in different human GBM cell lines. VIP receptor antagonists and a PACAP antibody enhanced migration, suggesting an autocrine or paracrine action of VIP-related peptides on this process [36,38]. The aim of the present study was to extend the analysis of GBM migration and invasion with the objective to investigate the signaling pathways involved in these processes. The C6 rat GBM cells were chosen because they have been long recognized as a valuable experimental model system for studying ex vivo or in vivo GBM progression after injection in Wistar rat brain [49]. This animal model was utilized in our group to test the in vivo antitumor activity of a cyclopamine glucuronide prodrug on GBM progression [50]. The U87 cell line has been demonstrated to express the PAC1 receptor and this cell line has been utilized to evaluate the effect of a VIP receptors antagonist on xenograft proliferation in nude mice [31]. We have generated C6 cells expressing the GFP protein to better follow the extent of tumor cell invasion after implantation of these cells inside the brain parenchyma. This allowed us to precisely determine the width and depth of invasion in rat brain slices by C6 cells and to demonstrate that both VIP and PACAP-38 were able to limit GBM cell invasion in this tissue. Of course, these ex vivo conditions do not reflect the real conditions of GBM development inside the brain. In the future, it will be of chief importance to study the effects of these neuropeptides on GBM progression on in vivo developing tumors in animal models. The wound healing technique was carried out to assess the efficiency of inhibitors of PKA (H89) and Akt (IA) protein kinases on the VIP and PACAP-38 inhibition of C6 and U87 migration process. The data indicated that the effects of these neuropeptides were abolished by these compounds, suggesting an involvement of the corresponding pathways in the migration process of GBM cells. In the C6 cells that are expressing important levels of VIP mRNA, the VIP receptors antagonist VIP_10-28_ increased cell migration, arguing for an autocrine or paracrine reduction by endogenous VIP of the C6 cells migration. The VIP_10-28_ antagonist has been demonstrated to be more selective for the VPAC1 and VPAC2 receptor subtypes [51,52], indicating their possible implication in the VIP and PACAP-38 inhibition of the C6 and U87 migration process. Interestingly, the fact that both VIP and PACAP-38 reduced U87 GBM migration also indicates that these peptides could act on the low amount of VPAC1 receptors expressed in this cell line or perhaps on certain splice variants of the PAC1 receptor that have been demonstrated to display high-affinity for both VIP and PACAP [53]. Increased migration by VIP_10-28_ was not observed in U87 cells, which may be due to a lesser expression of VIP, as observed by real-time RT-PCR analysis. In the C6 cells, H89, IA, GLI1 siRNA, and cyclopamine, were all capable to totally abolish the increased cell migration caused by VIP_10-28_. Migration was potently reduced in C6 transfected with a GLI1 siRNA and in these cells, VIP_10-28_ failed to increase cell migration. Taken together, these data argue for an implication of the PKA, PI3K/Akt, and SHH/GLI1 pathways in the migration process of GBM cells.

To confirm these hypotheses, effects of the neuropeptides were evaluated on nuclear GLI1 protein expression in C6 and U87 cells. The data indicated that VIP or PACAP-38 strongly reduced the nuclear level of GLI1 protein, a process which was abolished by the PKA inhibitor H89. In C6 cells, the VIP_10-28_ increase of nuclear GLI1 expression was completely blocked by IA, but the Akt inhibitor had no effect on VIP or PACAP-38 reduction of nuclear GLI1 level. These observations clearly demonstrate that regulation of GLI1 localization and activity by the VIP-receptor system are dependent on the PKA and the PI3K/Akt pathways. Regulation of the nuclear localization and transcriptional activity of GLI1 by Akt is well documented in a number of malignancies in general and in GBM in particular [54,55,56,57]. Furthermore, it has been described that phosphorylation of GLI1 near its nuclear localization signal (NLS) by PKA retains GLI1 in the cytoplasm [42].

A time-dependent treatment of C6 cells by PACAP-38 revealed that this neuropeptide could reduce the expression of the phosphorylated form of Akt with a very strong effect at 6 h. Conversely, PACAP-38 triggered a potent elevation of phospho-PTEN protein with the highest amplitude after 3 h of treatment, i.e., 3 h before the maximal inhibition of phospho-Akt, suggesting a close link between these two mechanisms. These experiments have been conducted in C6 cells because they express a wild-type PTEN protein, contrary to U87 presenting an in-frame deletion in the PTEN gene [58]. PKA could play a major role in the phosphorylation events observed in our studies because it has been shown that the p85 regulatory subunit of PI3K is phosphorylated in vivo and in vitro by PKA on its serine residue 83, a process which was demonstrated to play a major role on G1 cell cycle arrest [59]. Hence, the strong inhibition of Akt phosphorylation by PACAP-38 could be due to the combined phosphorylation by PKA of the P85 regulatory PI3K subunit and potentially of the phosphatase PTEN, leading to PIP3 dephosphorylation and to inhibition of the p110 catalytic PI3K subunit. The latter process remains, however, to be demonstrated in GBM cells. Subsequently, the decreased nuclear localization and activity of GLI1 observed in GBM cells could result both from the above discussed PKA-dependent inhibition of PI3K/Akt activity and from GLI1 phosphorylation by PKA (Figure 6). Hence PKA appears like a possible central core in the complex network linking these different pathways.

These data all together indicate that triggering the VIP-receptor system, a potent activator of the cAMP/PKA axis, reduces migration and invasion in GBM cells through a blockade of the PI3K/Akt and of the Shh/GLI1 pathways. These effects are efficiently abolished by H89, a PKA inhibitor. One has to keep in mind that pharmacological inhibitors can display off-target effects, as reported for H89, which besides PKA, can act on other protein kinases and signaling mechanisms [60,61]. In spite of these limitations, the observations reported in the present study support the hypothesis of a central role of PKA at a cross-road of signaling pathways involved in the regulation of GBM migration and invasion. Ten adenylate cyclases (ADCYs) have been identified in mammals. They can be activated by the plant diterpene Forskolin. Some of these isoforms are often overexpressed in brain cancers. ADCY3 is localized in the primary cilia of neural cells. Development of specific activators of certain ADCY isoforms expressed in GBM cells could thus be an interesting paradigm. Another strategy could be to block the phosphodiesterase PDE4 by Rolipram or other inhibitors of this enzyme. PDE4, which specifically hydrolyzes cAMP, is often overexpressed in cancer cells [43,44,45,46,47,48].

As a powerful activator of the cAMP/PKA pathway, the VIP-receptor system should also be considered as an alternative strategy to inhibit the interrelated signaling pathways involved in GBM cells in particular and in cancer progression in general. In this respect, a number of VIP or PACAP more stable derivatives or with improved receptor subtype specificity have been developed. Some of them have been utilized for in vivo studies on their neuroprotective properties. They really deserve to be considered for future in vitro and in vivo preclinical studies targeting the VIP-receptor system in cancer progression [62,63,64,65].

## 4. Materials and Methods

### 4.1. Materials

Synthetic VIP, PACAP-38, and antagonist VIP_10-28_ were purchased from Polypeptide Laboratories. H89 (a PKA antagonist) and IA (Akt Inhibitor VIII, Isozyme-Selective) were obtained from Calbiochem Merck, Guyancourt France. The Smo antagonist cyclopamine (an inhibitor of SHH pathway) was obtained from Sigma Aldrich, Saint-Quentin Fallavier, France).

### 4.2. Cell Culture

The rat C6 glioma cell line [66] was purchased from the European Collection of Animal Cell Culture (ECACC) (Salisbury, UK). The human U87 MG glioblastoma cell line was obtained from the American Type Culture Collection (Manassas, VA, USA). C6 cells were utilized from passage 9 to 20, to avoid the possible loss in these cells of efficient coupling of the VIP and PACAP receptors to adenylate cyclase at late passages reported by other investigators [33]. In standard monolayer conditions, the cells were cultured in Dulbecco’s modified Eagle’s medium (DMEM) with GlutamaxTM I, and supplemented with 10% fetal bovine serum, 100 U/mL penicillin, and 100 µg/mL streptomycin. Cells were incubated in humidified 95% air, 5% CO_2_ at 37 °C. Medium was changed twice a week, and cells were subcultured once a week using trypsin-EDTA solution. Cell culture media, supplements and reagents were from Lonza, Levallois-Perret, France.

### 4.3. Transfection with a GFP-Encoding Vector

C6 cells were stably transfected with copGFP plasmid (Santa Cruz Biotechnology, Clinisciences, Nanterre, France) using FuGENE (Promega, Charbonnières-les-Bains, France) according to the manufacturer’s instructions. Twenty-four hours before transfection experiments, 150,000 C6 cells were plated per well in 6-well dishes and grown in serum-containing medium without antibiotics. FuGENE (3 µL) and 1 µg of a copGFP plasmid was mixed with Opti-MEM and the mixture was incubated for 15 min at room temperature. The combined mixtures were added to the cells. After a 24-h culture period, the medium was changed and puromycin selection (12 µg/mL) of transfected cells was started.

### 4.4. siRNA Transfection

C6 cells were transfected with rat GLI-1 siRNA (Santa Cruz Biotechnology) using Lipofectamine RNAiMax (Invitrogen, Paris, France) according to the manufacturer’s instructions. Four microliters of Lipofectamine RNAiMax and 25 nM of GLI-1 siRNA or a control siRNA FITC Conjugate-A (Santa Cruz Biotechnology, Clinisciences, Nanterre, France) were each mixed with 100 µL of Opti-MEM (Invitrogen). The RNAiMax/Opti-MEM mixture was incubated for 5 min at room temperature and next combined with the vector/Opti-MEM mixture. The combined mixtures were incubated for 20 min at room temperature and then added to the cells. After a 24-h culture period, the medium was changed.

### 4.5. Ex Vivo Invasion Assay on Rat Brain Slices

The ex vivo invasion assay on rat brain slices was performed as described in Nakada and collaborators [67]. The experimental protocol was approved by the local Animal Care Committee (Comité d’éthique en expérimentation animale COMETHEA). Briefly, 400-μm-thick sections were prepared from newborn Wistar rat brain. Brain slices then were laid down on a Millicell-CM membrane insert (Millipore, Molsheim, France) and the insert was placed in individual wells of six-well plates. Medium (1 mL) was added to the bottom of culture plates. Slices were cultured in the same culture medium as C6 cells at 37 °C in a 5% CO_2_ incubator. After 24 h, a volume of 0.1 µL containing 5 × 10^3^ C6 cells stably expressing GFP protein was injected with a Hamilton syringe in the putamen of the brain slices. Two hours after injection, the culture medium was changed, and the slices were treated with 10^-6^M of VIP, PACAP-38, or VIP_10-28_. After 48 h of treatment, the slices were incubated for 4 h with 4% of paraformaldehyde. Imaging of specimens was performed using a macro-fluorescent imaging system (Olympus, Rungis, France) equipped with a GFP barrier filter (DP50, Olympus). The area of GFP-stained cells in invasive population (RIM) in each section was measured and the depth of invasion at 48 h was determined using a confocal FV-1000 station installed on an inverted microscope IX-81 (Olympus, Rungis, France).

### 4.6. Wound Healing

Cells were cultured in 24-well dishes and a wound was made in the confluent monolayer by mechanical scraping using a p200 pipette tip. After a wash with serum-free medium, the cells were incubated in the absence or presence of increasing concentrations of VIP, PACAP-38, or VIP_10-28_ in serum-free medium supplemented with 0.1% BSA. Cells underwent an incubation for 20 min with an inhibitor (1 µM of H89, IA, or cyclopamine) followed by a treatment with VIP, PACAP-38, or VIP_10-28_. Phase-contrast micrographs were collected after 24 h and the width of the wound was measured. The data in percent represented the wound healing closure according to the calculation: ((Initial width − final width)/initial width) × 100.

### 4.7. Western Immunoblotting and Antibodies

C6 and U87 cells were cultured in 25-cm^2^ flasks to about 75% confluence and then incubated for 0, 1/2, 1, 2, 3, 6, or 24 h, in the presence or absence of VIP or VIP_10-28_ 10^−7^ M in serum-free medium supplemented with 0.1% BSA. After treatments, cells were harvested with 1× trypsin/EDTA, washed twice with cold Phosphate Buffered Saline (PBS) and resuspended in 10 μL per 10^6^ cells of ice-cold lysis buffer (10 mM Tris, pH 7.5, 0.5 mM EDTA, pH 8.0, 0.5% CHAPS, 10% glycerol) supplemented with a cocktail of protease inhibitors (Calbiochem Merck, Guyancourt, France). After 30 min on ice, the samples were centrifuged at 4 °C for 20 min at 10,000× *g* and stored at −80 °C.

For extraction of nuclear and cytosolic proteins, cells were resuspended in 1 mL per 5 × 10^6^ cells of lysis buffer (10 mM Tris, pH 7.4, 3 mM MgCl_2_, 10 mM NaCl, 0.5% NP40). After 5 min on ice and centrifugation for 10 min at 2000× *g* at 4 °C, the supernatant containing the cytosolic fraction was collected. The pellet containing nuclei was resuspended in 30 μL per 5 × 10^6^ cells of lysis buffer (20 mM HEPES, pH 7.9, 0.3 M NaCl, 1.5 mM MgCl_2_, and the cocktail of protease inhibitors mentioned above) and incubated 30 min on ice under vigorous shaking. Samples were then centrifuged 20 min at 10,000× *g* at 4 °C. The supernatants were stored at −80 °C.

For all protein extracts, protein concentration was determined using the Bio-Rad DC Protein Assay (Bio-Rad, Marnes-La-Coquette, France). Total, cytosolic, and nuclear proteins were resolved in 8% SDS-PAGE and electroblotted for 1 h at 200 mA onto Immobilon-P membranes (Millipore). Membranes were blocked overnight at 4 °C using 5% nonfat milk in Tris-buffered saline (TBS) containing 0.1% Tween-20 (TBST) and incubated for 1 h at room temperature with a polyclonal rabbit anti-Akt antibody (1:1000; Cell Signaling Ozyme, San Quentin Yvelines, France), a polyclonal rabbit anti-phospho-Akt antibody (1:1000; Cell Signaling Ozyme, San Quentin Yvelines, France), a polyclonal rabbit anti-phospho-PTEN antibody (1:1000; Cell Signaling, Saint Quentin Yvelines, France), a polyclonal rabbit anti-Gli1 antibody (1:200; Santa-Cruz Clinisciences, Nanterre, France) or 1 h at room temperature with a monoclonal mouse anti-GAPDH antibody (1:80,000; Abcam, Paris, France) or monoclonal mouse anti-Lamin-C (1:1000; Abcam, Paris, France) used as a loading control, in blocking solution. After three washes, the membrane was next incubated 1 h at room temperature with goat anti-rabbit or goat anti-mouse secondary antibodies (1:20,000) (Calbiochem, Beeston Nottingham, UK) conjugated to horseradish peroxydase. The enhanced chemiluminescence (ECL) stainings obtained using ECL prime western blotting detection reagent (GE Healthcare, Buc, France) were quantified by densitometry with Image J software (National Institutes of Health, USA).

### 4.8. cDNA Synthesis

Total RNA was isolated using the GenElute_TM_ Mammalian Total RNA kit (Sigma-Aldrich, Saint-Quentin Fallavier, France) following the manufacturer’s instructions. Total RNA (1 µg) was treated with 1 U/µg RNA of DNase I Amplification Grade (Invitrogen) according to the manufacturer’s instructions, and in the presence of 10 U/µg RNA of RNaseOUT (Invitrogen). After DNase inactivation, RNA was reverse transcribed using random hexamers (Promega) and M-MLV Reverse Transcriptase H Minus (Promega) according to the manufacturer’s instructions.

### 4.9. Quantitative Polymerase Chain Reaction of Reverse Transcribed mRNA

Real-time polymerase chain reaction quantification was carried out with the LightCycler System (Roche Diagnostics, Basel, Switzerland) using the SYBR qPCR Premix Ex Taq (Takara, Saint-Germain-en-Laye, France). Sequences of primers used are listed in Appendix A. The cDNA amplification program was 10 s at 95 °C to activate ExTaq polymerase followed by at least 45 cycles of 5 s at 95 °C, 5 s at 60 °C, and 10 s at 72 °C.

## 5. Conclusions 

Triggering the VIP-receptor system reduces migration and invasion in GBM cells through a PKA-dependent blockade of the PI3K/Akt and of the SHH/GLI1 pathways. Therefore, the VIP-receptor system displays anti-oncogenic properties in GBM cells and PKA appears like a central core in this process. As a powerful activator of the cAMP/PKA pathway, targeting the VIP-receptor system should be considered as an alternative strategy to inhibit the interrelated signaling pathways involved in GBM and possibly other cancers progression.

## Figures and Tables

**Figure 1 cancers-11-00123-f001:**
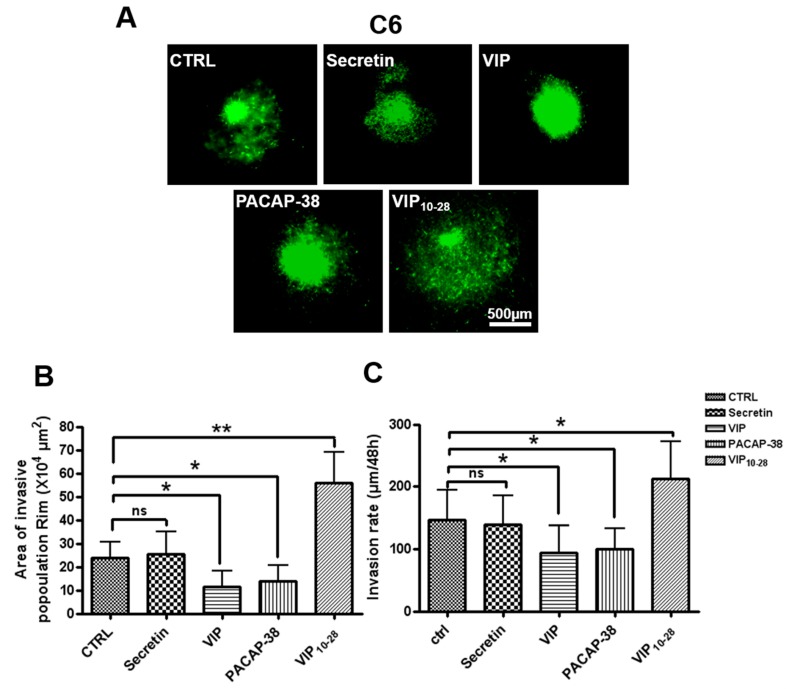
The rat brain slice model system was used to measure ex vivo C6 cell invasiveness. C6 cells stably expressing GFP were transplanted into the center of the putamen in brain slices. Then brain slices were treated or not (CTRL, control) with 10^-6^M of secretin, vasoactive intestinal peptide (VIP), PACAP-38, or VIP_10-28_, and the C6 cells were allowed to invade for 48 h. (**A**) Images were taken with a macroscope. (**B**) The area of GFP-stained cells in invasive population (RIM) in each section was measured. (**C**) The depth of invasion at 48 h was determined using the confocal microscopy. The values are the mean ± SD of quantification of glioblastoma (GBM) invasion in eight different brain slices, obtained from three independent experiments for each experimental condition. About 4–5 newborn rats were necessary to obtain 10 usable brain slices. Data were analyzed using the Kruskal–Wallis test and Dunn’s post hoc test (* *p* < 0.05; ** *p* < 0.01).

**Figure 2 cancers-11-00123-f002:**
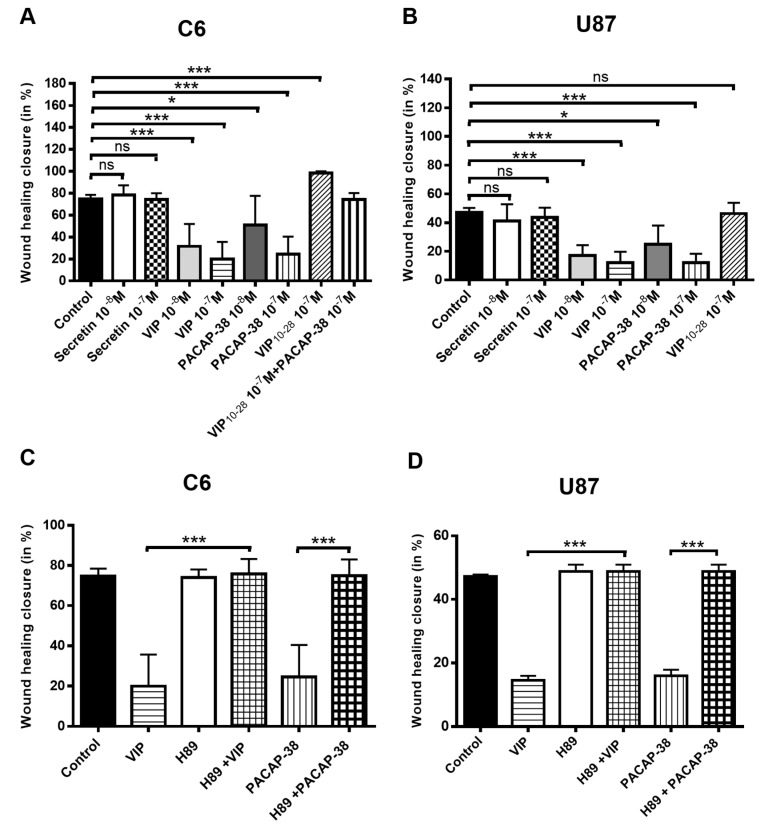
Effects of VIP, PACAP-38, VIP_10-28_ and/or inhibitors of signaling pathways H89 (PKA inhibitor), IA (Akt inhibitor), on the migration process of C6 and U87 cells. Migration was analyzed by the technique of wound healing closure, as described in Materials and Methods. (**A**,**B**) Effect of neuropeptides on the migration of C6 or U87 cells. (**C**,**D**) Migration of C6 and U87 cells treated or not with VIP or PACAP-38 at 10^−7^ M in absence or presence of H89. Data represent the wound healing closure expressed in % and are the mean ± SD of at least three independent experiments, each performed in duplicate. Data were analyzed using the Kruskal–Wallis test and Dunn’s post hoc test (* *p* < 0.05, *** *p* < 0.001, ns: not significant).

**Figure 3 cancers-11-00123-f003:**
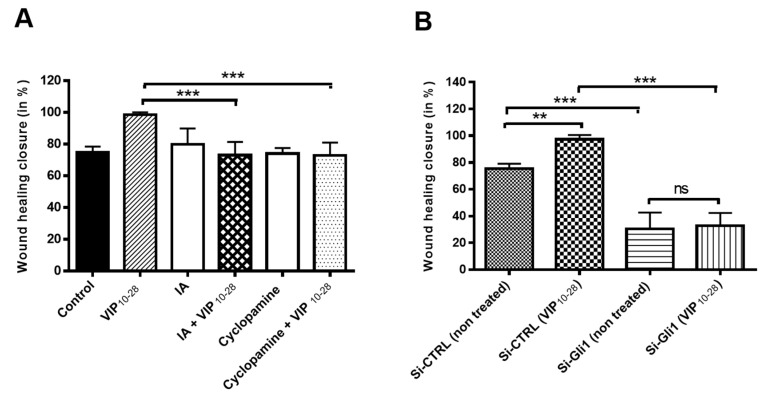
Effects of VIP_10-28_ and/or inhibitors IA (Akt inhibitor), cyclopamine (Smo inhibitor), or Gli1 (Si-RNA) on the migration process of C6 cells. (**A**) Migration of C6 treated or not with VIP_10-28_ at 10^−7^ M in absence or presence of 1 µM of IA or cyclopamine. Cells underwent incubation for 20 min with an inhibitor (1 µM of IA or cyclopamine) followed by treatment with VIP_10-28_ at 10^−7^ M for 24 h. (**B**) Wound healing assay using confluent C6 cells transfected with 25 nM siRNA targeting GLI1 (GLI1-siRNA) or 25 nM negative control siRNA. Cells were wounded and treated or not with VIP_10-28_ 10^−7^ M for 24 h. Data represent the wound healing closure expressed in % and are the mean ± SD of at least three independent experiments, each performed in duplicate. Data were analyzed using the Kruskal–Wallis test and Dunn’s post hoc test (** *p* < 0.01, and *** *p* < 0.001, ns: not significant).

**Figure 4 cancers-11-00123-f004:**
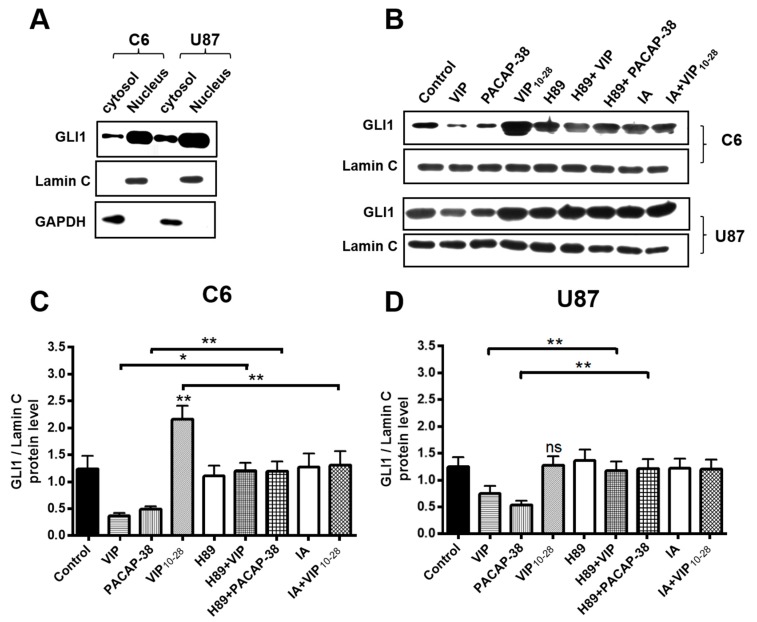
Effects of VIP, PACAP-38, VIP_10-28_, and/or inhibitors H89 (PKA inhibitor), IA (Akt inhibitor) on the expression and intracellular localization of GLI1 protein in C6 and U87 cells. (**A**) Western immunoblotting of GLI1 proteins in C6 and U87 cells. Cytosolic and nuclear proteins of cells were resolved by SDS-PAGE, transferred to membranes, and further probed with antibody anti-GLI1, anti-GAPDH, or anti-Lamin-C. GAPDH and Lamin-C proteins are used as a control for cytosolic and nuclear extracts, respectively. (**B**) Western immunoblotting of GLI1 proteins in C6 and U87 cells treated or not with neuropeptides in absence or presence of pathways inhibitors. Nuclear proteins were extracted from cells treated or not with neuropeptides at 10^−7^ M for 24 h. Cells were incubated for 20 min with an inhibitor (1 µM of H89 or IA) followed by treatment with VIP, PACAP-38, or VIP_10-28_ at 10^−7^ M for 24 h. Nuclear proteins of cells were resolved by SDS-PAGE, transferred to membranes and further probed with antibody anti-GLI1 or anti-Lamin-C. (**C**,**D**) Cumulative data from three independent experiments, in same conditions as in (**B**), after densitometry quantification of immunoblotting (WB) data. Quantification data are the mean ± SD. Data were analyzed using the Kruskal–Wallis test and Dunn’s post hoc test (* *p* < 0.05 and ** *p* < 0.01, ns: not significant).

**Figure 5 cancers-11-00123-f005:**
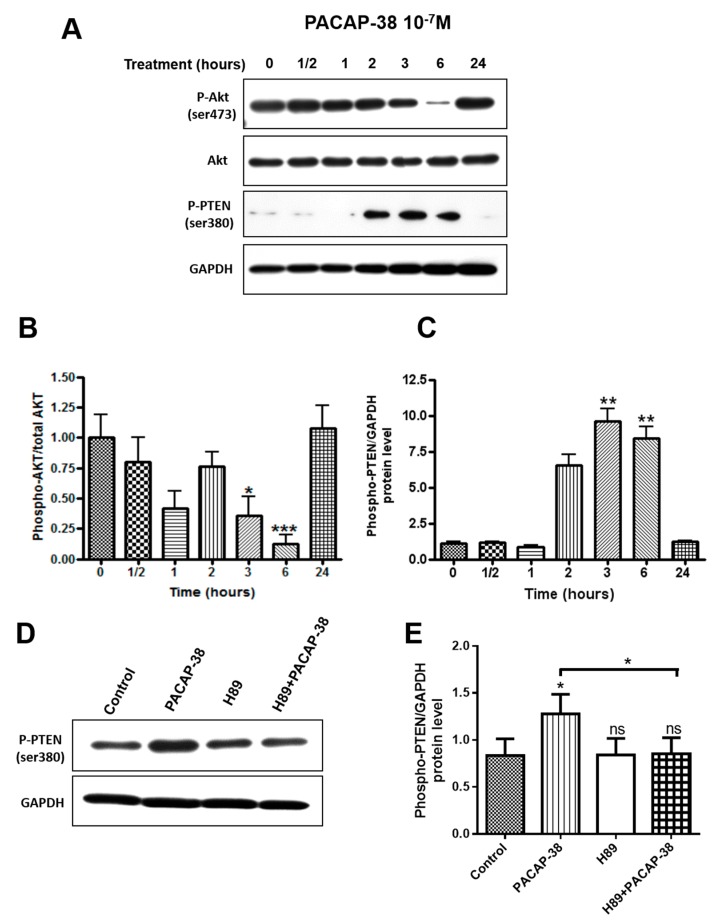
Effect of PACAP-38 on the phosphorylation of Akt and PTEN in C6 cells. Effect of the H89 PKA inhibitor on PACAP638 induced inhibition of PTEN phosphorylation. (**A**–**C**) Cells were treated with 10^−7^ M PACAP-38 for the indicated times. Total cell lysates were analyzed by immunoblotting (WB) with antibodies recognizing total Akt, phosphorylated Akt, phosphorylated PTEN or GAPDH. (**A**) corresponds to a representative WB experiment. (**B**,**C**) Cumulative data from three independent experiments after quantification by densitometry. Data were normalized by attributing a value of 1 for the control experiment. (**D**,**E**) Cells were treated for 3 h with 10^−7^ M PACAP-38, in the presence or not of 1 µM H89. (**D**) corresponds to a representative WB experiment. (**E**) Cumulative data from three independent experiments after quantification by densitometry. Quantification data are the mean ± SD. Data were analyzed using the Kruskal–Wallis test and Dunn’s post hoc test (* *p* < 0.05, ** *p* < 0.01, and *** *p* < 0.001, ns: not significant).

**Figure 6 cancers-11-00123-f006:**
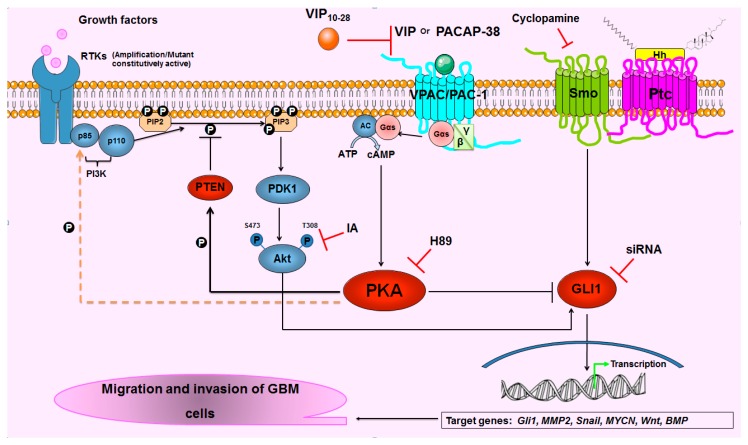
Potential molecular pathways and mechanisms regulated by the VIP-receptor system in glioblastoma (GBM) cells. Binding of VIP and related peptide agonists to their G-protein coupled receptors (GPCR) VPAC-1/2 and PAC1, leads to adenylate cyclase (AC) and cAMP-dependent Protein Kinase (PKA) activation. PKA is a central enzyme in these mechanisms because it could potentially: (i) inhibit glioma-associated oncogene homolog 1 (GLI1) nuclear translocation and activity; (ii) phosphorylate the phosphatase tensin-homolog deleted on chromosome ten (PTEN) and activate its Phosphatidylinositol (3,4,5)-trisphosphate (PIP3) phosphatase activity, leading to an inhibition of Akt phosphorylation and activation and consequently to a reduced nuclear localization and transcriptional activity of GLI1; (iii) phosphorylate the p85 regulatory subunit of PI3K on its serine residue 83. This hypothetic process (dashed line) that was described in vitro and in vivo to play a major role on G1 cell cycle arrest (see the discussion section) remains however to be demonstrated in GBM. Hh: Sonic Hedgehog; Ptc: Patched, the Hh receptor; RTK: receptor tyrosine kinase; Smo: smoothened, the Ptc co-receptor.

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
