# Peer review of "PKA at a Cross-Road of Signaling Pathways Involved in the Regulation of Glioblastoma Migration and Invasion by the Neuropeptides VIP and PACAP"

_cancers, 2019, doi:10.3390/cancers11010123_

Reviewer 1 Report

The manuscript by Bensalma and Colleagues describes an interesting series of experiments conducted in vitro to demonstrate that the effect of VIP signalling in modulating glioblastoma migration and invasion is mediated by cAMP-dependent protein kinases. 

I see two main limitations of this work, which may be overcome when designing future experiments:

this is an in-vitro approach, with only a short-term experiment on migration of cells in explanted brain slices: this limits the validity of inferences that may be drawn, since in 50 hours (the duration of the experiment from cell injection to fixation) no tumor structure (including vessels etc) can be formed. This limitation must be clearly added in the discussion. Studying the effects on in-vivo formed tumors would be the next necessary step. 

the demonstration that VIP acts via PKA is mainly based on the use of H-89 (fig 2 C and D, line 252...), which is a non-specific inhibitor of PKA, meaning that it has many other effects on other kinases (see for example Lochner and Moolman Cardiovasc Drug Rev. 2006 Fall-Winter;24(3-4):261-74). This must be stated in the discussion.

Besides these comments, I suggest some other corrections that should be made to the text:

line 46: it is not clear that GBM may arise from 'normal' neural stem cells or GBM stem cells. Please smooth the phrase.

line 100: GBM is not a neural tumor cell. The references to medulloblastoma may be confusing since the origin of GBM and MB is different. Please make clearer or omit MB.

Statistics, throughout the text: Kruskal-Wallis test says that there is a difference among different medians: what post-hoc test was used to check for differences between two values (as depicted in the bar graphs), after having done the Kruskal-Wallis?

line 166: 'en'

line 181: '-7' superscript

Fig 4 C and D: use the same scale for consistency

Line 191: 'Lamine' ==> 'Lamin'

Fig 5: I have some problems in comparing parts A with B and C. The vertical axis in B is the ratio phospho-AKT/total AKT, which means the first row of  panel A divided the second row of A: the WB in A says that all bands from 0 to 2 hours pf phospho-AKT are larger than total AKT, but there is no column in B larger than 1. Also, is P-PTEN at 2,3, 6 hours really 7-10 times larger than GAPDH, as the legend would say (Phospho-PTEN/ GAPDH protein level) ? I see them smaller, in A. Please explain. 

line 266: 'oberrved'

Graphical abstract legend: if the legend should be published, it would be nice to add the abbreviation.

Methods: it is mandatory to add the passage number at which the cells were used in all experiments, in order to replicate them properly, since cells may change phenotype during passaging.

line 332: 'new born rat Wistar brain' => 'newborn Wistar rat brain'

line 393: please state the entire PCR program (including first and last cycles and number of cycles).

In conclusion, it is a nice piece of in vitro work, suggestive of interesting development, but whose functional meaning in the tumor remains to be demonstrated.

Author Response

The manuscript by Bensalma and Colleagues describes an interesting series of experiments conducted in vitro to demonstrate that the effect of VIP signalling in modulating glioblastoma migration and invasion is mediated by cAMP-dependent protein kinases. 

I see two main limitations of this work, which may be overcome when designing future experiments:

this is an in-vitro approach, with only a short-term experiment on migration of cells in explanted brain slices: this limits the validity of inferences that may be drawn, since in 50 hours (the duration of the experiment from cell injection to fixation) no tumor structure (including vessels etc) can be formed. This limitation must be clearly added in the discussion. Studying the effects on in-vivo formed tumors would be the next necessary step.

This comment is now added in the discussion. Line 249.

The demonstration that VIP acts via PKA is mainly based on the use of H-89 (fig 2 C and D, line 252...), which is a non-specific inhibitor of PKA, meaning that it has many other effects on other kinases (see for example Lochner and Moolman Cardiovasc Drug Rev. 2006 Fall-Winter;24(3-4):261-74). This must be stated in the discussion.

This comment on the limitations of H-89 specificity is now added in the discussion and references cited in the bibliography.

Line 301.

Besides these comments, I suggest some other corrections that should be made to the text:

line 46: it is not clear that GBM may arise from 'normal' neural stem cells or GBM stem cells. Please smooth the phrase.

The phrase has been smoothened, see line 46

line 100: GBM is not a neural tumor cell. The references to medulloblastoma may be confusing since the origin of GBM and MB is different. Please make clearer or omit MB.

Neural has been removed. The repression of GLI1 expression and activity in a PKA-dependent manner has been historically demonstrated in the medulloblastoma and it really needs to be cited here.

Statistics, throughout the text: Kruskal-Wallis test says that there is a difference among different medians: what post-hoc test was used to check for differences between two values (as depicted in the bar graphs), after having done the Kruskal-Wallis?

line 166: 'en' corrected

line 181: '-7' superscript corrected

Fig 4 C and D: use the same scale for consistency. Figure has been rebuilt in this respect.

Line 191: 'Lamine' ==> 'Lamin' corrected

Fig 5: I have some problems in comparing parts A with B and C. The vertical axis in B is the ratio phospho-AKT/total AKT, which means the first row of  panel A divided the second row of A: the WB in A says that all bands from 0 to 2 hours pf phospho-AKT are larger than total AKT, but there is no column in B larger than 1. Also, is P-PTEN at 2,3, 6 hours really 7-10 times larger than GAPDH, as the legend would say (Phospho-PTEN/ GAPDH protein level) ? I see them smaller, in A. Please explain.

The normalization of the control value to 1 was wrong in this figure. The values have been recalculated and the figure rebuilt in this respect.

line 266: 'oberrved' corrected

Graphical abstract legend: if the legend should be published, it would be nice to add the abbreviation.

Abbreviations have been added.

Methods: it is mandatory to add the passage number at which the cells were used in all experiments, in order to replicate them properly, since cells may change phenotype during passaging.

Passages have been added together with a comment and and important reference concerning the loss of coupling of VIP/PACAP receptors to adenylate-cyclase at late passages. line 341 and ref 33

line 332: 'new born rat Wistar brain' => 'newborn Wistar rat brain'  corrected

line 393: please state the entire PCR program (including first and last cycles and number of cycles).

This has been precised, line 428

In conclusion, it is a nice piece of in vitro work, suggestive of interesting development, but whose functional meaning in the tumor remains to be demonstrated.

Thanks a lot, we appreciate your interest for our work

Reviewer 2 Report

Bensalma et al. carried out analyses of GBM migration and invasion with the aim to study the signaling pathways involved in these processes (PKA, VIP). The work is well written, assays correctly selected and described, results and discussion well presented, however some points should be addressed:

1) I suggest to add info on immunotherapy for glioblastoma (e.g. discussion part) as clinical trials test innovative immunotherapy approaches against brain tumors (e.g. NK cells, Adaptive T cell therapy, check point inhibitors, and their combinations). Also references should reflect latest immunotherapy approaches.

2) Indeed, PI3K/Akt, Sonic Hedgehog (SHH) and PKA 17 pathways play major regulatory roles in the progression of GBM. Also mentioning of  other e.g. mTOR, pRB, PDGF/PDGFR  pathways can be more informative and broaden to content, making the message more comprehensive  for potential readers.

3) Ex vivo assay on rat brain samples (slides) has been carried out. Here I would like to know more bout the number of slides, and number of mice tested in described settings. Additionally, in vivo study would be very welcome to supplement ad strength the research and observed findings.

Author Response

Bensalma et al. carried out analyses of GBM migration and invasion with the aim to study the signaling pathways involved in these processes (PKA, VIP). The work is well written, assays correctly selected and described, results and discussion well presented, however some points should be addressed:

1) I suggest to add info on immunotherapy for glioblastoma (e.g. discussion part) as clinical trials test innovative immunotherapy approaches against brain tumors (e.g. NK cells, Adaptive T cell therapy, check point inhibitors, and their combinations). Also references should reflect latest immunotherapy approaches.

A review on immunotherapy trials has been cited in the introduction (line 59).

2) Indeed, PI3K/Akt, Sonic Hedgehog (SHH) and PKA 17 pathways play major regulatory roles in the progression of GBM. Also mentioning of  other e.g. mTOR, pRB, PDGF/PDGFR  pathways can be more informative and broaden to content, making the message more comprehensive  for potential readers.

mTOR has been added to the cascade PI3K/Akt/mTOR (line 63). Sorry for the pathways which have not been cited in the introduction but we have tried to focus on mechanisms which gave rise to the development of therapeutic strategies and clinical trials against GBM.

3) Ex vivo assay on rat brain samples (slides) has been carried out. Here I would like to know more bout the number of slides, and number of mice tested in described settings. Additionally, in vivo study would be very welcome to supplement ad strength the research and observed findings.

The numbers of slices and mice tested have been added in the figure 1 legend, line 143.

Reviewer 3 Report

Dear authors,

This elegant study by Bensalma et al describes the molecular mechanisms implicated on VIP/PACAP system effects on glioblastoma migration and invasiveness. This is in line with previous studies from this team. The methodologies used are adequate and the results are very robust and well presented. Two minor comments are as follows:

Concerning the results with VIP or PACAP treatments on U87 on the wound healing model (Figure 2B), it is previously reported that these cells express mainly the PAC1 receptor, which is selective for PACAP. Nevertheless, borth peptides seem to exhibit the same effect at the same concentrations. What is the explanation for this unexpected result?

What VIP/PACAP receptor is targeted by VIP10-28, and why was this molecule chosen?

Author Response

Dear authors,

This elegant study by Bensalma et al describes the molecular mechanisms implicated on VIP/PACAP system effects on glioblastoma migration and invasiveness. This is in line with previous studies from this team. The methodologies used are adequate and the results are very robust and well presented. Two minor comments are as follows:

Thanks a lot for these encouraging comments on our work. We appreciate.

Concerning the results with VIP or PACAP treatments on U87 on the wound healing model (Figure 2B), it is previously reported that these cells express mainly the PAC1 receptor, which is selective for PACAP. Nevertheless, borth peptides seem to exhibit the same effect at the same concentrations. What is the explanation for this unexpected result?

This is a very good question and comments and references have been added in the duscussion in this respect. See from line 261.

What VIP/PACAP receptor is targeted by VIP10-28, and why was this molecule chosen?

This is a very good question and comments and references have been added in the duscussion in this respect. See from line 263.